# Sparsity regularization via tree-structured environments for disentangled representations

**Elliot Layne** *elliot.layne@mail.mcgill.ca*
*McGill University, Mila*

**Jason Hartford** *jason@valencelabs.com*
*Valence Labs*

**Sébastien Lachapelle** *lachaseb@mila.quebec*
*Université de Montréal, Mila*
*Samsung - SAIT AI Lab, Montreal*

**Mathieu Blanchette** *mathieu.blanchette@mcgill.ca*
*McGill University, Mila*

**Dhanya Sridhar** *dhanya.sridhar@mila.quebec*
*Université de Montréal, Mila*

**Reviewed on OpenReview:** *https://openreview.net/forum?id=ZzUz0jo200*

## Abstract

Many causal systems such as biological processes in cells can only be observed indirectly via measurements, such as gene expression. Causal representation learning—the task of correctly mapping low-level observations to latent causal variables—could advance scientific understanding by enabling inference of latent variables such as pathway activation. In this paper, we develop methods for inferring latent variables from multiple related datasets (environments) and tasks. As a running example, we consider the task of predicting a phenotype from gene expression, where we often collect data from multiple cell types or organisms that are related in known ways. The key insight is that the mapping from latent variables driven by gene expression to the phenotype of interest changes sparsely across closely related environments. To model sparse changes, we introduce Tree-Based Regularization (TBR), an objective that minimizes both prediction error and regularizes closely related environments to learn similar predictors. We prove that under assumptions about the degree of sparse changes, TBR identifies the true latent variables up to some simple transformations. We evaluate the theory empirically with both simulations and ground-truth gene expression data. We find that TBR recovers the latent causal variables better than related methods across these settings, even under settings that violate some assumptions of the theory.

## 1 Introduction

Discovering new knowledge using machine learning is made ever more possible with the growing amounts of unstructured data that we collect, from images, video, and text to experimental measurements from large-scale biological assays. However, scientific discovery from such low-level measurements requires representation learning, the task of mapping low-level observations to a high-level feature space.

As a running example, consider learning about drivers of disease from gene expression measurements, where genes drive protein abundance levels that mediate disease. If we want to discover meaningful features that could be causally linked to outcomes, we need methods to learn a *disentangled* representation. This can be

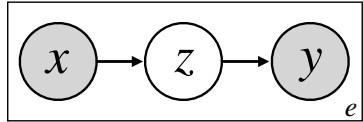

Figure 1: The causal directed acyclic graph (DAG) relating observations $\mathbf{X}$, latents $\mathbf{Z}$ and target variable of interest $Y$. The marginal of $\mathbf{X}$ and $\mathbf{Z} \to Y$ relationship is specific to environment $e$.

done an encoder that maps observations to causally relevant latent variables, and not to transformations of these variables that change their meaning.

Disentanglement is impossible from independent and identically distributed (IID) data without assumptions to constrain the solution space (Hyvärinen & Pajunen, 1999; Locatello et al., 2019), but by leveraging known structure such as non-stationarity (Hyvarinen & Morioka, 2017; Hyvarinen et al., 2019) or sparsity (Brehmer et al., 2022; Ahuja et al., 2022c; Lachapelle et al., 2022b), it is possible to identify latent variables. The challenge is in finding assumptions that are strong enough to rule out spurious solutions, while remaining flexible enough to fit the domain of interest. To this end, this paper proposes Tree-Based Regularization (TBR), a new disentangled representation learning method that is particularly well-suited to biological settings. TBR leverages non-stationarity from multiple datasets – called environments – that are hierarchically related via a known tree.

In our running biology example, the environments are defined by cell types, and their relations come from the tree describing the cell-type differentiation process Enver et al. (2005). We assume that the relationship between low-level observations and latent variables, represented by $P(\mathbf{Z}|\mathbf{X})$, is constant across these environments as it is driven by the underlying physics of the cell. However, the mechanism driving the outcome (e.g., disease), $P_e(Y|\mathbf{Z})$, changes across environments, reflecting processes like variation across cell types or evolution.

The key assumption we make is that across closely related environments, the effects of only a sparse set of true latents will vary. That is, changes to the conditional distributions $P_e(Y|\mathbf{Z})$ are sparse, such that environments that are siblings in the known tree share the same functional form up to a few parameters, while more distant environments will have accumulated more changes. TBR enforces this structure via a sparsity-inducing penalty. These constraints reflect biological settings, where it is common to observe data across multiple cell types or model organisms with hierarchical relationship structures (e.g. a phylogenetic tree), and allow us to learn representations that identify the true latent variables up to irrelevant transformations.

We show both theoretically and empirically that by enforcing this sparsity assumption on changes of the conditional distributions, it is possible to identify the ground truth latent variables up to a permutation and scaling factor, assuming the presence of a sufficient number of environments such that the effects of each variable are varied. We further analyze the sensitivity of TBR to assumption violations and find that it remains robust up to a point. Then, we apply TBR to a dataset of gene expression measurements across different cell types, where we hold out some genes as latent variables and simulate phenotypes derived from these latents. We find that TBR recovers the true latent variables with higher fidelity than standard representation learning approaches, which then leads to higher transfer learning performance, demonstrating a concrete benefit of the TBR approach.

## 2 Problem Formulation

We consider an input vector $\mathbf{X}$ (e.g., gene expression) and a target variable of interest $Y$ (e.g., disease phenotype). We observe $n$ iid samples $(\mathbf{x}_i^e, y_i^e)_{i=1}^n$ from $E$ different environments where each $e$ denotes the environment that produced the sample. Figure 1 illustrates the causal model of the data generating process, which we note is distinct from most causal representation learning works as $Z$ is a causal descendant of $X$, rather than a causal ancestor. The inputs $\mathbf{X}$ drive latent features $\mathbf{Z}$ that mediate the effect on $Y$. $P(Y|\mathbf{Z})$ varies across environments while $P(\mathbf{Z}|\mathbf{X})$ remains invariant. Specifically, we consider the following generative

process:

$$\mathbf{Z} = \Psi(\mathbf{X}) + \eta; \quad Y^e = w_e^\top \mathbf{Z}^e + \epsilon \tag{1}$$

There exists an environment-invariant continuous function $\Psi$ relating $\mathbf{X}$ to $\mathbf{Z}$, with noise term $\eta$, assumed to be within the space of functions approximable with a neural network (Cybenko, 1989). Additionally, there exists an environment-specific function $\Phi_e$ that relates latent factors $\mathbf{Z}^e$ to $Y^e$. In particular, we consider the case where $\Phi_e$ is a linear mapping with weights $w_e \in \mathbb{R}^k$ where $k = |\mathbf{Z}|$, and $\epsilon$ is an environment-independent noise term. Linearity of the output map is clearly restrictive, but given that $\Phi(\cdot)$ can be a basis function expansion of $\mathbf{Z}$, this process could still capture a variety of nonlinear maps between latents and targets.

Critically, we posit that the environments have diverged according to some evolution-like process. We denote directed tree $\mathcal{T} = (\mathcal{E}, \mathcal{A})$, where environments $\mathcal{E}$ are the nodes of the tree, and $\mathcal{A} \subset \mathcal{E}^2$ denotes the set of arcs[1] between environments. The environment $0 \in \mathcal{E}$ is assumed to be the root environment. In practice, we will often observe samples only from leaf environments, and not from internal nodes in the tree (which represent historical environments, such as ancestral species or transient progenitor cells), but our framework is agnostic to this.

We assume that the environment-specific parameters $w_e$ are subject to mutations between parent-child environment pairs. In addition, we assume that these mutations are **sparse**, such that across each arc $a = (e_m, e_n)$, there exists a subset of indices $\mathcal{S} \subset \{1, ...k\}$ such that $\forall i \in \mathcal{S}, w_{e_m}[i] \neq w_{e_n}[i]$, and the components of $w_{e_m}$ and $w_{e_n}$ are equal otherwise. We denote the vector of differences in parameters across arc $a$ as $\delta_a = w_{e_n} - w_{e_m}$. Note that the weights for any environment $w_e$ can be expressed as the sum of the root weights $w_0$ and the relevant mutation vectors:

$$w_e := w_0 + \sum_{a \in \mathrm{path}(0,e)} \delta_a \tag{2}$$

where $\mathrm{path}(0, e)$ denotes the set of arcs forming the path from the root environment 0 to environment $e$. We denote $\Delta \in \mathbb{R}^{|\mathcal{A}| \times k}$ as the matrix whose rows are the set of vectors $\delta_a$ for each arc $a \in \mathcal{A}$.

Figure 2 illustrates an example data-generating process involving a tree with 5 environments and 3 leaves.

The proposed evolution-like process relating environments via sparse local changes causes the distributions $P^e(Y \mid \mathbf{Z})$ to gradually change. This model of evolution aligns with many observed processes in nature where sparse genetic or epigenetic changes alter the process leading to a given phenotype, such as cell type differentiation and species evolution.

## 2.1 Tree-based regularization

We approximate $\Psi$ with an arbitrary deep neural network $\hat{\Psi}_\theta$ with parameters $\theta$:

$$\hat{\mathbf{Z}} = \hat{\Psi}_\theta(\mathbf{X}) \tag{3}$$

We estimate a set of weights $\hat{w}_0 \in \mathbb{R}^k$, along with a matrix $\hat{\Delta} \in \mathbb{R}^{|\mathcal{A}| \times k}$, whose rows contains the estimates $\hat{\delta}_a$ for each $a \in \mathcal{A}$, permitting estimation of each $\hat{w}_e$ in the form of Eq. 2.

We jointly optimize $\hat{\Psi}_\theta, \hat{w}_0$ and $\hat{\Delta}$ to produce optimal predictions $\hat{Y}_e$ across all environments. Additionally, we regularize our estimate $\hat{\Delta}$ with a sparsity-inducing norm $||\hat{\Delta}||_0$.

Assuming that we only observe data samples in a subset of environments $\mathcal{L} \subset \mathcal{E}$, typically the leaves in $\mathcal{T}$, this yields the following loss function:

$$\mathrm{Loss} = \sum_{e \in \mathcal{L}} (w_e^\top \hat{\Psi}_\theta(\mathbf{X}^e) - Y^e)^2 + \lambda ||\hat{\Delta}||_0 \tag{4}$$

where $\lambda > 0$ is a scaling parameter. The first term in Eq. 4 is a standard prediction error term, minimizing the mean-squared error (MSE) for predictions of $\hat{Y}$. The second term regularizes the $L_0$ norm of the matrix $\Delta$.

---

[1]An *arc* is a directed edge between two nodes.

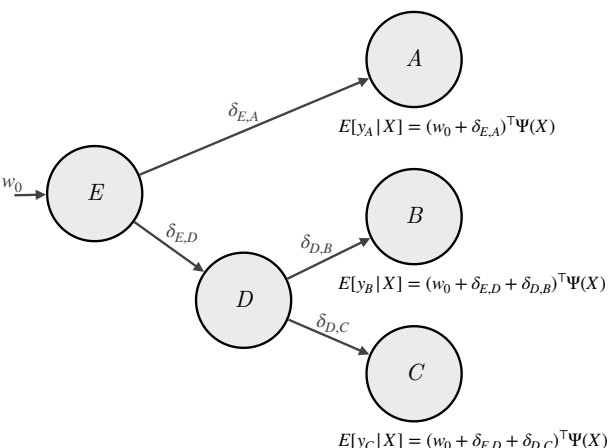

Figure 2: An example of posited DGP and corresponding TBR parameterization for a dataset with samples originating from three different environments ($A, B$, and $C$). A set of weights $W_0$ is associated to the root. A trainable update $W_e$ is associated to each edge $e$. The prediction function $f_u$ at node $u$ will make predictions on data samples originating from environment $u$. Adapted from Layne et al. (2020).

Since parameters in $\Delta$ are the differences across each arc in $\mathcal{A}$, by maximizing the sparsity of $\Delta$, we encourage parent-child environment pairs to learn similar predictors. This captures the essence of our motivation, that the mechanisms controlling $P(Y \mid \mathbf{Z})$ should only change rarely. We will denote our method, optimizing Eq. 4, as *tree-based regularization* (TBR).

## 3 Identifiability via Sparse Tree-Based Regularization

This section presents our main theoretical contribution on disentanglement. More formally, we will show that the learned $\hat{\mathbf{Z}}$ identifies the true latent variables $\mathbf{Z}$ up to permutation and element-wise rescaling, i.e., $\hat{\mathbf{Z}} = \mathbf{DPZ}$ where $\mathbf{D}$ is an invertible diagonal matrix and $\mathbf{P}$ is a permutation. It is necessary that the dimension $|\hat{Z}| = |Z|$. We discuss estimation of $|Z|$, should it be unknown, in § 4.

The problem of non-identifiability arises because the first term in the loss function in Eq. 4, which penalizes prediction errors, admits many solutions for $\hat{\mathbf{Z}}$. Consider the generative process specific to environment A in Figure 2:

$$Y = (w_0 + \delta_A)^\top \mathbf{Z} + \epsilon \tag{5}$$

We can see that the distribution of $Y$ remains unchanged if $\mathbf{Z}$ is subject to some linear transformation $\mathbf{L}$, so long as an inverse transformation is applied to $w_0$ and $\Delta$:

$$Y = (w_0^\top \mathbf{L}^{-1} + \delta_A^\top \mathbf{L}^{-1})\mathbf{LZ} + \epsilon \tag{6}$$

However, the solution $\hat{\mathbf{Z}} = \mathbf{LZ}$ potentially linearly entangles the components of the true features $\mathbf{Z}$. For tasks that require the learned features to be semantically faithful to the true features, e.g., inferring causal effects of the features, this entanglement will lead to biased conclusions.

The second term in the loss in Eq. 4 plays a key role in selecting disentangled solutions among all solutions that achieve optimal prediction error. This is because the regularization term $\lambda||\hat{\Delta}||_0$ is generally not invariant to linearly entangled solutions such as $\mathbf{LZ}$, as because for most choices of $\mathbf{L}$, $||\Delta||_0 \neq ||\Delta\mathbf{L}^{-1}||_0$, unless $\mathbf{L} = \mathbf{PD}$ (when the representation is disentangled).

The core of the theory in this section lies in establishing assumptions under which the only linear transformations that can be applied to $\Delta$ while maintaining the same level of sparsity, and thus the regularization penalty in Eq. 4, are permutation and scaling operations. Combined with a result demonstrating the identifiability of $\mathbf{Z}$ up to linear transformation, this regularization constraint permits disentanglement of both $\Delta$ and the latent features $\mathbf{Z}$.

Our intuition focused on linearly entangled solutions $\hat{\mathbf{Z}} = \mathbf{L}\mathbf{Z}$ but in general, the hidden layer of a neural network could produce arbitrarily entangled solutions. As such, we prove the identifiability of features learned using tree-based regularization in two parts. First, we establish conditions under which the features learned with a nonlinear function, e.g., a neural network, are identified up to linear transformations, bringing us back to the linearly entangled case. This result builds on proof techniques found in the literature (Khemakhem et al., 2020b;a; Roeder et al., 2021; Ahuja et al., 2022a; Lachapelle et al., 2022a).

Second, we show that with some assumptions, the sparsity constraint on $||\hat{\Delta}||_0$ permits further identification of $\mathbf{Z}$ up to permutation and scaling. Specifically, we show that any $\Delta$ with at most one non-zero value per row cannot be entangled without a resulting increase in regularization cost.

## 3.1 Linear Identification of Z

**Assumption 3.1** (Data-generating process (DGP)). For each environment $e$, we assume the pairs $(x, y)$ are drawn i.i.d from a distribution satisfying

$$\mathbb{E}[Y \mid \mathbf{X}, e] = w_e^\top \Psi(\mathbf{X}). \tag{7}$$

Additionally, we assume the support of $p(x \mid e)$, denoted by $\mathcal{X} \in \mathbb{R}^d$, is fixed across environments.

**Assumption 3.2** (Sufficient task variability). We assume that there exist environments $e_1, \ldots, e_k \in \mathcal{E}$ such that the matrix $[w_{e_1}, \ldots w_{e_k}]$ is invertible.

**Assumption 3.3** (Sufficient representation variability). We assume there exist $x^1, \ldots, x^k \in \mathcal{X}$ such that the matrix $[\Psi(x^1), \ldots, \Psi(x^k)]$ is invertible.

Taken together, these assumptions establish our first requirement, by implying that the latents are identified up to a linear transformation $\mathbf{L}$,

**Proposition 3.4.** *Suppose Assumptions 3.1, 3.2 & 3.3 hold. Moreover, consider the learned parameters $\hat{w}_0$ and $\{\hat{\delta}_a\}_{a \in \mathcal{A}}$ and the learned encoder function $\hat{\Psi}(x)$. Analogously to Eq. 2, we define $\hat{w}_e := \hat{w}_0 + \sum_{a \in path(0,e)} \hat{\delta}_a$ for all $e \in \mathcal{E}$. If for all $\mathbf{X} \in \mathcal{X}$ and all $e \in \mathcal{E}$ we have $\mathbb{E}[Y|\mathbf{X}, e] = \hat{w}_e^\top \hat{\Psi}(\mathbf{X})$, then, there exists an invertible matrix $\mathbf{L} \in \mathbb{R}^{k \times k}$ such that*

1. *For all $x \in \mathcal{X}$, $\Psi(x) = \mathbf{L}\hat{\Psi}(x)$;*

2. *For all $e \in \mathcal{E}$, $w_e^\top \mathbf{L} = \hat{w}_e^\top$; and*

3. *For all $a \in \mathcal{A}$, $\delta_a^\top \mathbf{L} = \hat{\delta}_a^\top$.*

The proof is included in Appendix A.1.

### Identification Up To Scaling and Permutation

The previous section shows that we can identify every $\delta_a$ up to some common invertible linear transformation $\mathbf{L}$. Recall $\Delta, \hat{\Delta} \in \mathbb{R}^{|\mathcal{A}| \times k}$ are simply the concatenations of the $\delta_a$ and $\hat{\delta}_a$, respectively. This means we can write $\Delta \mathbf{L} = \hat{\Delta}$. We now show that penalizing the $L_0$-norm of our estimate $\hat{\Delta}$ allows us to identify $\Delta$ up to a permutation and scaling.

**Assumption 3.5** (1-sparse perturbations). Each $\delta_a$ has at most one nonzero value.

**Assumption 3.6** (Sufficient perturbations). For all $i \in \{1, \ldots, k\}$, there exists $a \in \mathcal{A}$ such that $\delta_{a,i} \neq 0$.

**Proposition 3.7** (Disentanglement via 1-sparse perturbations). *Suppose Assumptions 3.5 & 3.6 hold, let $\mathbf{L}$ be an invertible matrix and let $\hat{\Delta} := \Delta \mathbf{L}$. If $||\hat{\Delta}||_0 \leq ||\Delta||_0$, then $\mathbf{L}$ is a permutation-scaling matrix.*

**Proof sketch** To understand the crux of proving this result, recall the aspect of TBR that drives the result in the first place: the $L_0$ norm prefers solutions $\hat{\Delta}$ that are sparse. For this regularization to choose the disentangled solution, $\hat{\mathbf{Z}} = \mathbf{Z}\mathbf{D}\mathbf{P}$, we have to show that $\hat{\Delta} = \mathbf{Z}^\top \mathbf{L}^{-1}$, the result of linearly entangled solution,

cannot be sparser than $\hat{\Delta}$ for the disentangled solution. To this end, our strategy will be to fix the amount of sparsity in the ground-truth matrix $\Delta$. Then, we will show by contradiction that there exists a mapping between the columns of $\Delta$ and $\hat{\Delta}$ such that each column in $\Delta$ is at least as sparse as the corresponding column in $\hat{\Delta}$. Using this result, we then show that to meet our constraint on the sparsity of $\hat{\Delta}$, it must be the case that $\mathbf{L}$ is a permutation-scaling matrix.

While Assumption 3.5, which requires all rows in $\Delta$ to have at most 1 non-zero entry, is unlikely to hold in many practical applications of interest, we do note that we empirically consider violations of this assumption in § 4, and find the results promising. Some further discussion about relaxation of the 1-sparse assumption is included in Appendix A.2.

### 3.2 Proof of Proposition 3.7

This result builds on arguments from both Freyaldenhoven (2020) and Lachapelle et al. (2022b).

*Proof.* By Lemma A.1, there exists a permutation $\pi$ such that for each column $i$ in $\mathbf{L}$, there is an index $\pi(i)$ where $\mathbf{L}_{\pi(i),i}$ is a non-zero value $\alpha$. Thus $\hat{\Delta}_{:,i}$ is equal to some linear combination of columns in $\Delta$ where column $\Delta_{:,\pi(i)}$ has coefficient $\alpha$. Details are included in Appendix A.

We use operator $S$ to denote the "sparsity pattern" of a matrix. The sparsity pattern of $\Delta$ is denoted as $S_\Delta$, and is equal to the set of indices of non-zero elements in $\Delta$. $S_\Delta^c$ indicates the complement of the sparsity pattern: the set of indices corresponding to entries with value zero. Similarly, $S_{\Delta_{:,i}}$ denotes the sparsity pattern of $\Delta$ at column $i$.

Our proof builds upon a series of column-wise comparisons. For each index $i \in \{1, \ldots, k\}$, we compare the $L_0$ norm of column $\Delta_{:,\pi(i)}$ and column $\hat{\Delta}_{:,i}$.

We define two sets of indices, $\Omega_i$ and $\Gamma_i$,

$$\Omega_i := S_{\Delta_{:,\pi(i)}} \cap S_{\hat{\Delta}_{:,i}}^c \tag{8}$$

$$\Gamma_i := S_{\Delta_{:,\pi(i)}}^c \cap S_{\hat{\Delta}_{:,i}} \tag{9}$$

Thus, $\Omega_i$ represents nonzero entries that were present in $\Delta_{:,\pi(i)}$, but not in $\hat{\Delta}_{:,i}$ and $\Gamma_i$ represents the opposite, i.e. nonzero entries that were not present in $\Delta_{:,\pi(i)}$ but that are present in $\hat{\Delta}_{:,i}$. Thus, the following equality holds:

$$||\hat{\Delta}_{:,i}||_0 = ||\Delta_{:,\pi(i)}||_0 + |\Gamma_i| - |\Omega_i| \tag{10}$$

We now show that $\Omega_i$ is empty by contradiction. Assume there exists $a \in \Omega_i$. This means $\Delta_{a,\pi(i)} \neq 0$ and $\hat{\Delta}_{a,i} = \Delta_{a,:}L_{:,i} = 0$. By the 1-sparse assumption, $\Delta_{a,-\pi(i)} = 0$. This further implies that $\Delta_{a,:}L_{:,i} = \Delta_{a,\pi(i)}L_{\pi(i),i} = 0$, which is a contradiction. We thus conclude that $\Omega_i$ is empty.

Therefore, for arbitrary column $i$, it will always be the case that $||\hat{\Delta}_{:,i}||_0 \geq ||\Delta_{:,\pi(i)}||_0$.

We rewrite the original sparsity constraint:

$$||\hat{\Delta}||_0 \leq ||\Delta||_0 \tag{11}$$

$$\sum_i ||\hat{\Delta}_{:,i}||_0 \leq \sum_i ||\Delta_{:,i}||_0 \tag{12}$$

$$\sum_i ||\hat{\Delta}_{:,i}||_0 \leq \sum_i ||\Delta_{:,\pi(i)}||_0 \tag{13}$$

$$\sum_i (||\hat{\Delta}_{:,i}||_0 - ||\Delta_{:,\pi(i)}||_0) \leq 0 \tag{14}$$

The above combined with $||\hat{\Delta}_{:,i}||_0 - ||\Delta_{:,\pi(i)}||_0 \geq 0$ implies that, for all $i$, $||\hat{\Delta}_{:,i}||_0 = ||\Delta_{:,\pi(i)}||_0$. This means $\Gamma_i$ is empty as well.

Because $\Gamma_i$ is empty for all $i$, we have that

$$\forall i, a, \ \Delta_{a,\pi(i)} = 0 \implies \hat{\Delta}_{a,i} = 0, \tag{15}$$

or, equivalently,

$$\forall i, a, \ \Delta_{a,i} = 0 \implies \hat{\Delta}_{a,\pi^{-1}(i)} = 0, \tag{16}$$

Choose an arbitrary $i$. By Assumption 3.5 & 3.6, there exists an $a$ such that $\Delta_{a,\pi(i)} \neq 0$ and $\Delta_{a,-\pi(i)} = 0$. By Eq. 16, we have

$$\Delta_{a,-\pi(i)} = 0 \implies \hat{\Delta}_{a,\pi^{-1}(-\pi(i))} = 0 \tag{17}$$

$$\hat{\Delta}_{a,-i} = 0 \tag{18}$$

$$\Delta_{a,:}\mathbf{L}_{:,-i} = 0 \tag{19}$$

$$\Delta_{a,\pi(i)}\mathbf{L}_{\pi(i),-i} = 0, \tag{20}$$

which implies $\mathbf{L}_{\pi(i),-i} = 0$ (since $\Delta_{a,\pi(i)} \neq 0$). This holds for all $i$, thus $\mathbf{L}$ is a permutation-scaling matrix.

$\square$

## 4 Empirical Studies

The goal of the empirical studies is to demonstrate that TBR recovers disentangled latents, which further lead to accurate causal inference and transfer learning. To that end, we conduct studies with simulated data and gene expression data across cell types to investigate the behaviour of TBR under various generative settings, robustness to violations of Assumption 3.5 , and potential downstream benefits to disentangled representations. For all these experiments, we compare the representations produced by TBR to those from a baseline method: a standard linear map $w^*$ fit to the target label $Y$ on top of the learned nonlinear representation $\hat{\Psi}$, yielding the following optimization objective:

$$\text{Loss}_{\text{baseline}} = \sum_{e \in \mathcal{L}} (w^{*\top}\hat{\Psi}_\theta(\mathbf{X}^e) - Y^e)^2 \tag{21}$$

We note that throughout experimentation, the regularization of $||\hat{\Delta}||_0$ was approximated via regularization of $||\hat{\Delta}||_1$, following the practice of other sparsity-based disentanglement methods (Lachapelle et al., 2022a;b). This approximation introduces a slight bias into our model by shrinking the non-zero values of $||\hat{\Delta}||$. Additional implementation details are included in Appendix A.5. All results are averaged over 10 repetitions unless otherwise specified. Additionally, in the simulation study, we consider a second baseline, which we refer to as the "Sparse Predictors" method, following the approach developed by Lachapelle et al. (2022b). This method learns predictors for multiple tasks that share a representation $\hat{Z}$ such that each task $Y$ depends sparsely on the components of the true underlying latents $Z$. We implement this method by training environment-specific weights $w_e$ with $L_1$ regularization to induce sparsity. Although Lachapelle et al. (2022b) show that this form of sparsity regularization suffices to recover disentangled features $\hat{Z}$, the sparse dependence assumption does not hold in our settings, where *mechanism changes*– not mappings –are sparse. As such, we hypothesize that the Sparse Predictors approach will not produce disentangled features in our studies.

**Performance metric** In Figure 3a, we evaluate the disentanglement of $\hat{\mathbf{Z}}$ using the mean correlation coefficient (MCC). The MCC between $\mathbf{Z}$ and $\hat{\mathbf{Z}}$ is defined by:

$$\arg\max_{\pi \in \mathcal{P}} \frac{1}{k} \sum_{i=1}^{k} || \text{Pearson}((\hat{\mathbf{Z}}_{\pi(i)}), \mathbf{Z}_i) ||, \tag{22}$$

where $\mathcal{P}$ is the set of possible permutations of $\hat{\mathbf{Z}}$ and we calculate the Pearson correlation coefficient.

An MCC score of 1.0 indicates an estimated representation is equal to the ground truth up to permutation and scaling.

**Simulation details** The general simulation procedure is: we generate a balanced binary tree $\mathcal{T}$, of depth 7 with a total of 128 leaves. In each non-root environment, we sample 3000 datapoints according to,

$$
\begin{aligned}
\mathbf{x} \in \mathbb{R}^{16} &\sim \mathcal{N}(0, 1) \\
\mathbf{z} \in \mathbb{R}^{5} &\sim \mathcal{N}(\Phi(\mathbf{x}), 0.1) \\
\Phi(\mathbf{x}) &= \tanh(\mathbf{w}^{\top}\mathbf{x}); \quad \mathbf{w} \sim \mathcal{N}(0, 1) \\
w_0 &\sim \mathcal{N}(0, 1)
\end{aligned}
\tag{23}
$$

Each arc $a$ in $\mathcal{T}$ is assigned a sparse $\delta_a \in \mathbb{R}^5$, with non-zero Gaussian values ($\sigma^2 = 0.25$) at a random subset of $S$ indices, where $S$ denotes the number of non-zero entries and is varied throughout experimentation. $w_e$ is generated using Equation 2, and the target $Y$ is sampled according to Equation 1, with $\epsilon \sim \mathcal{N}(0, 0.1)$. In Appendix A.5, we show results for an equivalent simulation where only leaf nodes are observed, more closely matching biological applications.

**Disentanglement.** To study the first question about validating our disentanglement theory, we evaluated the MCC of the representations learned by TBR and the baseline methods while varying the level of sparsity in $\Delta$, from $S = 0$, indicating that each $\delta \in \Delta$ is a null vector, to $S = 5$, resulting in a fully dense $\Delta$. The results, displayed in Figure 3a, align with our theoretical findings. With $S = 0$, $\Delta$ is completely sparse, and $\hat{\Psi}$ is free to entangle $\hat{\mathbf{Z}}$ without any increase in regularization cost. In this setting, TBR, "Sparse Predictors" and the baseline methods each learn entangled representations , exhibiting mean MCC of approximately 0.5.

In the $S = 1$ setting, which uniquely satisfies Assumption 3.5, TBR elicits mean MCC scores of $0.98 \pm 0.01$ indicating that the representations of $\hat{\mathbf{Z}}$ are almost fully disentangled. "Sparse Predictors" exhibits mean MCC scores of $0.81 \pm 0.12$, and the baseline method achieves mean MCC of $0.47 \pm 0.06$

**Sensitivity to assumption violations.** The settings of $S \geq 2$ test the robustness of our method to violations of Assumption 3.5, which require 1-sparse changes across environments. As $S$ increases, we see that the MCC scores obtained by TBR gradually decline, a reasonable result given that the guarantees of § 3 no longer hold. However, TBR remains notably stronger than the baseline methods with $S <= 4$, indicating that tree-based regularization incentivizes partial disentanglement even in settings not covered by our theory. Some additional results exploring further variations on our simulation setting are included in Appendix A.5, showing robustness to all-linear DGPs, more complex $X \rightarrow Z$ relationships, varied distributions of $X$ and more stochastic amounts of variation in $\Delta$.

**Predictive performance.** We evaluate the MSE of $\hat{Y}$ for each method while varying the sparsity level $S$ for each $\delta \in \Delta$. Figure 3b highlights these results. When $S = 0$, $\mathbb{E}[Y|Z]$ is invariant across environments, matching the model fit by the standard baseline as well as TBR and Sparse Predictors. As expected, all methods perform well in this case. As $S$ increases, TBR and Sparse Predictors maintain low prediction error, while the performance of the baseline method quickly degrades. This aligns with expectations, as the baseline method lacks the capacity to fit the environment-specific effects of the latents $\mathbf{Z}$.

**Prediction error and estimating latent dimension.** We further find evidence that varying the dimension of $|\hat{\mathbf{Z}}|$ in order to minimize predictive error can enable estimation of the true dimension of $|\mathbf{Z}|$ if it is unknown, satisfying a key assumption of our method. We measured the MSE of $\hat{Y}$ achieved by the baseline and TBR when varying dimensionality of $\hat{\mathbf{Z}}$ from 1 to 8, while setting the average sparsity to half the dimensionality by setting each entry in $\Delta$ to zero with probability 0.5. For both methods, and for four different values of the dimension $|\mathbf{Z}|$, setting $|\hat{\mathbf{Z}}| \leq |\mathbf{Z}|$ resulted in elevated MSE. Notably, the MSE achieved by TBR distinctly plateaued after reaching the correct dimensionality, as is shown in Figure 4. While not conclusive, these results suggest that there is indeed potential in the analysis of prediction error with TBR as a method of estimating the number of latent features present.

**Causal effect estimation.** We estimated the causal effects of the high-level latent variables $\mathbf{Z}$ on the target $Y$ using the representation $\hat{\mathbf{Z}}$ produced by each method as a substitute for the true latents. The goal is to investigate whether disentangled representation learning is necessary for tasks such as causal inference where the semantics of the variables involved matter. Formally, we estimate the average treatment effect

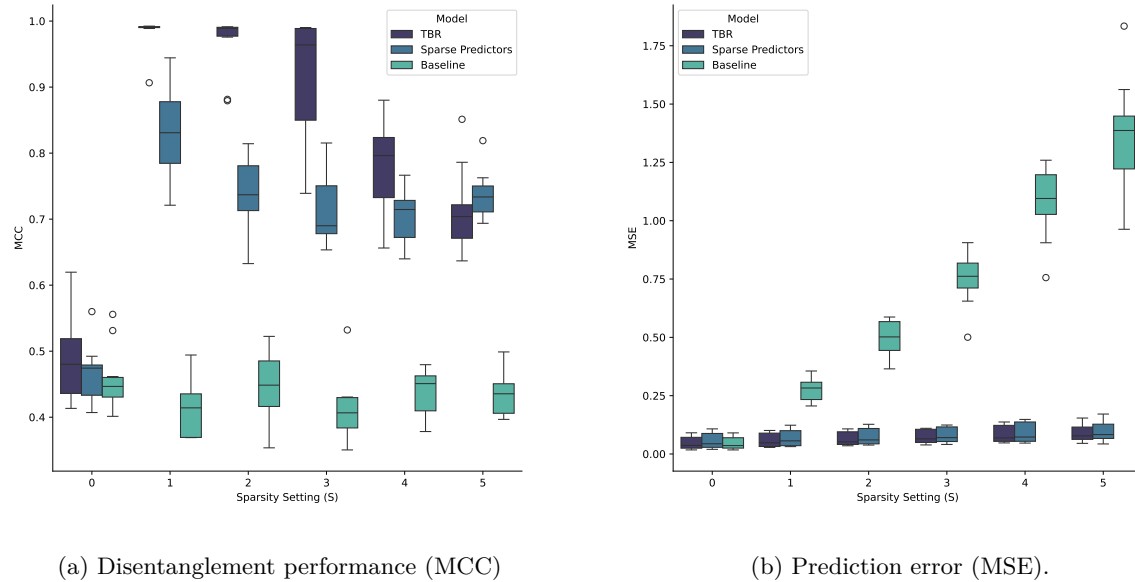

(a) Disentanglement performance (MCC)

(b) Prediction error (MSE).

Figure 3: Assessment of model performance across various simulation settings. The x-axis indicates the number of non-zero entries in each $\delta$. **Left:** A comparison of the MCC between $\mathbf{Z}$ and the estimates $\hat{\mathbf{Z}}$ produced by TBR and the baseline methods. TBR achieves near perfect disentanglement when $S = 1$. **Right:** Comparison of prediction error produced by TBR and the baseline methods when estimating $Y$.

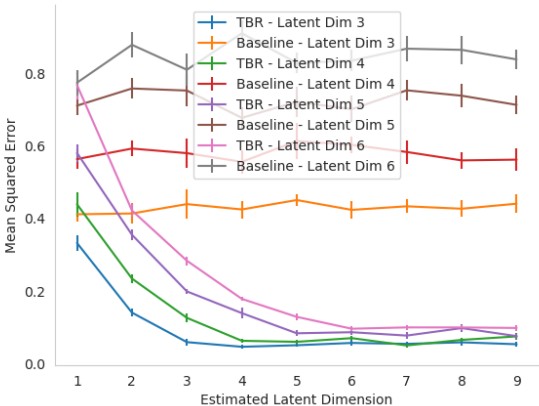

Figure 4: An analysis on the effects of changing the dimension of $\hat{\mathbf{Z}}$ on MSE, across various dimensionalities of $\mathbf{Z}$. For $|\mathbf{Z}| \in \{4, 5, 6\}$, TBR achieved minimal predictive loss when $|\hat{\mathbf{Z}}| = |\mathbf{Z}|$. In contrast, the baseline method had no cases where the global minimum for MSE occurred when the estimated and true latent vectors were of the same dimension.

(ATE) of each latent $\mathbf{Z}_k$ on $Y$,

$$\text{ATE}_k = \mathbb{E}[Y; \text{do}(\mathbf{Z}_k = z + \delta)] - \mathbb{E}[Y; \text{do}(\mathbf{Z}_k = z)]$$

which measures the change in the mean of $Y$ when we intervene (indicated by the do operator) and increase the value of $\mathbf{Z}_k$ by some constant $\delta$. Since each ATE is not identified from observational data without assumptions, we follow the necessary assumptions that: (i) there are no hidden confounding variables affect

both $\mathbf{Z}$ and $Y$ and (ii) for all values $\tilde{z}$ of $\mathbf{Z}_{-k}, 0 < P(\mathbf{Z}_k = z | \mathbf{Z}_{-k} = \tilde{z}) < 1$. Following Pearl et al. (2000), with these assumptions, and from the graphical model in Figure 1, each causal effect is identified by,

$$\text{ATE}_k = \mathbb{E}_{\mathbf{Z}_{-k}}\left[\mathbb{E}[Y | \mathbf{Z}_{-k}, \mathbf{Z}_k = z + \delta] - \mathbb{E}[Y | \mathbf{Z}_{-k}, \mathbf{Z}_k = z]\right]$$

As a consequence of the linearity in the mapping from $\mathbf{Z}$ to $Y$, each ATE can be estimated by including all variables $\mathbf{Z}$ in a linear regression of $Y$ and reading off the fitted linear coefficients.

In our experiments, we estimate $\text{ATE}_k$ in ten randomly selected environments in the 1-sparse setting. We fit ordinary least squares regression to $Y$ using the representations $\hat{\mathbf{Z}}$ learned using TBR and the baseline methods and compared the estimated effects to those estimated when using the ground truth $\mathbf{Z}$. We normalized the representation values and calculated the mean squared error of the absolute coefficient values, to ignore the effects of scaling. Table 3, included in the appendix, shows the results from this study. We see that the disentangled representation obtained by TBR yields highly accurate effects (MSE=$0.05 \pm 0.07$) in this setting while the baseline method produces representations that severely introduce bias into the effect estimates (MSE=$1.51 \pm 1.8$). Further details are presented in Appendix A.6.

## 4.1 Single-cell RNA-seq experimentation

In addition to analyzing the behavior of TBR in simulation, we evaluated the ability of our method to learn disentangled representations of ground-truth gene expression data, with simulated downstream cell-state phenotypes.

In our setup, both $\mathbf{X}$ and $\mathbf{Z}$ consisted of real gene expression data observed across different cells. We selected genes from within a set of 11 broadly expressed house-keeping genes(Eisenberg & Levanon, 2013) to hold out as latent variables. For $\mathbf{X}$, we extracted all transcription factors (TFs) annotated as regulators of $\mathbf{Z}$, in the hTFtarget database (Zhang et al., 2020). We removed from consideration two weakly regulated house-keeping genes, resulting in a set of nine candidate $\mathbf{Z}$ genes (see Appendix A.5), each regulated by 18 to 95 TFs within $\mathbf{X}$. We performed all experimentation on the publicly available GTEx V8 snRNA-Seq dataset, described by Eraslan et al. (2022). We consider the 43 epithelial cell types as the set of environments $\mathcal{E}$.

**Preprocessing** All preprocessing of the GTEx data was performed using standard functions within the `Scanpy` library (Wolf et al., 2018). To denoise the data and adjust for dropout during sequencing, we first applied across the entire dataset the MAGIC function developed in Van Dijk et al. (2018). The resulting dataset was highly imbalanced, with a small number of cell types comprising the majority of examples. To prevent these cell types from dominating our analysis, we randomly selected a subset of each cell-type population such that no individual environment had more than 1000 samples, resulting in a final dataset of 15256 cells.

We constructed the tree $\mathcal{T}$ relating the cell types by running the dendrogram method in the `Scanpy` library, such that the 43 epithelial cell types made up the leaves in $\mathcal{T}$. We simulated phenotype $Y$ using latent gene expression $\mathbf{Z}$ and cell type tree $\mathcal{T}$ via the simulation procedure outlined in § 4.

**Results** We repeatedly sampled subsets of 5 candidate genes to serve as $\mathbf{Z}$ and the corresponding TF expression $\mathbf{X}$, and compared the performance of TBR and our previously described baseline method across 30 different simulated phenotypes, for each sparsity level $S$.

The performance of the two methods in disentangling the latent gene expression values are depicted in Figure 5a. TBR and the baseline method exhibit equivalent average MCC scores of $0.51 \pm 0.01$ and $0.50 \pm 0.01$ respectively in the $S = 0$ setting, when there is no variation between environments. Once variation is introduced, the performance of TBR increases, exhibiting an average MCC of $0.61 \pm 0.02$ with both $S = 1$ and $S = 2$, while the performance of the baseline method degrades rapidly to $0.31 \pm 0.02$ as $K$ increases to 2. Globally, MCC scores are lower than observed in simulation. We expect that much of this gap in performance can be attributed to the high levels of endogenous noise within the $\mathbf{X} \rightarrow \mathbf{Z}$ relationship.

**Generalization to unseen environments.** Since parameter values across related environments only sparsely vary when we consider the true latents, but densely vary when we consider incorrect transformations

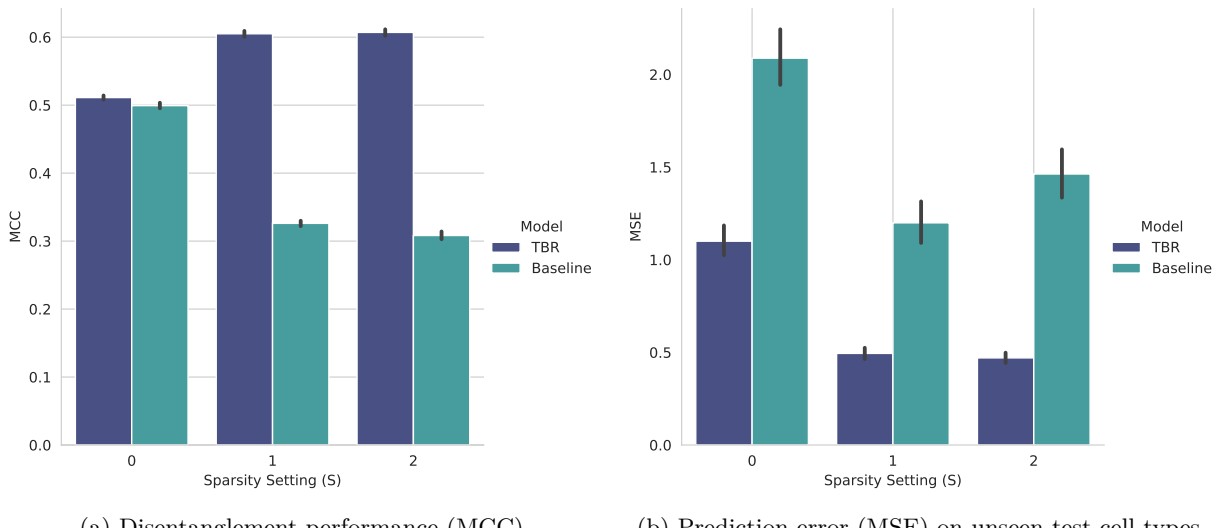

(a) Disentanglement performance (MCC)

(b) Prediction error (MSE) on unseen test cell types.

Figure 5: **Left panel:** A comparison of the MCC between $\hat{\mathbf{Z}}$ and $\mathbf{Z}$ across various settings of sparsity in the simulation procedure, when $\mathbf{X}$ and $\mathbf{Z}$ are both ground-truth gene expression measurements. Results are averaged over 30 simulated phenotypes for each of 75 random choices of genes to serve as $\mathbf{Z}$. **Right panel:** MSE exhibited when generalizing trained instances of TBR and the baseline method to unseen cell types with 1-Sparse generative parameters changes, while varying the $S$ setting of training environments. Results are averaged over 10 simulated phenotypes each for 50 random choices of $\mathbf{Z}$ and 4 held-out test cell types.

of these latents, we further expect that better disentanglement should lead to lower error when we transfer our learned predictor to an unseen but similar environment. Thus, to demonstrate the value of learning partially disentangled representations of gene expression, we evaluated the ability of our methods to predict phenotype in environments that were unobserved during training. We selected cell types to use as a test set, and completely excluded them from training.

We compared the performance between our baseline method, trained on the closest neighboring cell-type to the test type, and TBR, using the parameters from the closest environment to the test type in $\mathcal{T}$. To maximize the potential performance from the baseline, we only considered test cell types whose closest neighbor had a large population (pop. size $>= 1000$). Additionally, we restricted analysis to test types with a population size $>= 100$.

As in the previous experiment, we repeatedly sampled random choices of genes to make up $\mathbf{Z}$ and generated simulated phenotypes with varying settings of $S$. In all cases, the unobserved environments had $w_{test}$ exhibiting 1-sparse changes from the closest observed environment in $\mathcal{T}$, resulting in a non-trivial generalization challenge even when $S = 0$ for the training environments. The results averaged across all choices of $\mathbf{Z}$ are presented in Figure 5b.

Notably, training TBR with $S = 0$ resulted in a test MSE of $1.10\pm0.58$ in the unseen cell types, a significantly higher error rate than the $S = 1$ and $S = 2$ settings (respective MSE of $0.49 \pm 0.22$ and $0.47 \pm 0.20$). This improvement in MSE when $S > 0$ is expected, as the 1-sparse changes in the unseen $w_{test}$ could become more widely distributed across the entangled estimated of $\hat{\mathbf{Z}}$. This reduction in generalization risk in the $S >= 1$ setting illustrates the value of even partially disentangled representations. The baseline method exhibits significantly higher MSE in all settings, which may be due to the inherent limitations of training on only closely related cell types, restricting sample size.

## 5 Related work

Several papers have shown that with samples from multiple environments, auxiliary labels, and assumptions on the data generating process, we can learn disentangled representations (Hyvarinen & Morioka, 2016; 2017; Hyvarinen et al., 2019; Khemakhem et al., 2020a;b; Von Kugelgen et al., 2021; Shen et al., 2021; Klindt et al., 2021; Gresele et al., 2019). Most of these previous works assume the latent variables are causal ancestors of the observed variables, whereas we consider latent variables which are causal descendants of the observations, enabling use of the target distribution as a signal for latent identification, rather than a reconstruction objective. Within the research on disentanglement, this paper relates most closely to two lines of work that exploit sparsity to guarantee disentangled solutions to the representation learning task.

**Sparse changes to latents.** One line of work focuses on temporal observations (Lippe et al., 2022) or paired samples (Locatello et al., 2019; Ahuja et al., 2022b; Von Kugelgen et al., 2021) generated by varying only a small number of generative factors. Similarly, Lachapelle et al. (2022b; 2024) leverage sparse interactions between latent factors across time and/or with auxiliary variables to learn disentangled representations. Instead of sparse latent differences between paired samples or units across time, recent works Ahuja et al. (2023); Buchholz et al. (2023); Squires et al. (2023); Zhang et al. (2023); Varici et al. (2024); von Kügelgen et al. (2023) consider unpaired data from multiple environments that sparsely differ due to single-node interventions to the latent generative factors. In this paper, instead of leveraging sparse shifts in the distribution over latent variables $P(\mathbf{Z})$, we exploit sparse changes in $P(Y|\mathbf{Z})$, the mechanism by which the latent variables affect a target variable.

**Sparse mappings.** Another line of work exploits sparsity in the mappings from latent to observed variables to establish disentanglement. In the case of generative factors, Moran et al. (2022); Brady et al. (2023); Zheng et al. (2022) assume that each latent factor affects a subset of the components of the observation. In the latent features setting that we also study in this paper, Lachapelle et al. (2022a); Fumero et al. (2023) observe multiple targets of interest that depend sparsely on the latent features. In contrast, in this paper, we focus on sparse changes in the mapping from latent features to the single target variable of interest. Another key distinction is that we leverage a hierarchical structure over environments. This allows us to enforce that mechanism changes be sparse locally while still admitting changes that are dense between environments that have diverged significantly. Thus we do not need to constrain all pairs of environments to vary sparsely.

**Tree based regularization** Layne et al. (2020) employ regularization scheme similar to TBR, with the goal of improving generalization performance when modelling phylogenetically distributed datasets. Layne et al. (2020) leveraged the assumption distributional shifts of the features or labels, and applied $L_2$ regularization to changes in parameters across neighboring nodes in a phylogenetic tree in order to improve predictive performance. In contrast, in this work we explicitly assume that there are sparse changes in the effects of latent variables, and show how regularizing sparsity across neighboring environments can be used to learn disentangled representations of these latents.

## 6 Conclusion

The ability to learn disentangled representations is an important prerequisite for the use of representation learning in a number of scientific tasks, such as causal effect estimation. There is a growing attention in the literature to the use of sparse variations between environments to achieve disentanglement. We make several novel contributions towards this line of pursuit. We demonstrate that disentanglement is possible when data originates from a set of evolving environments, such as those resulting from cellular differentiation or species evolution, where parameter shifts along a given tree branch are sparse. Tree-branch sparsity suffices to enable disentanglement, even if the resulting pair-wise differences between observable environments may be dense.

We present novel theoretical results, leveraging sparse tree-based regularization in parameter space to disentangle representations of learned features, downstream from the observable inputs in the causal graph. We validate the value of our approach through a series of simulation studies, that demonstrate improvements in disentanglement scores, prediction error, and estimation of the causal effects of the latent features. We further investigate the ability of TBR to learn disentangled representations of ground truth gene expres-

sion data, with simulated downstream cell phenotypes. While further improvements are required to achieve the full potential of causal representation learning on scRNA-seq data, we observe promising trends. Our findings suggest that there is value in leveraging information about cell-type differentiation with TBR-style methodology, both to assist in disentangling latent molecular mechanisms and to improve generalization performance.

There are multiple directions for future work, including better characterization of the number of environments required, further exploration into estimation of the number of latents, and expanding our results into more flexible generative processes, such as by relaxing the assumption that $p(\mathbf{Z} \mid \mathbf{X})$ remains invariant.

A future direction of particular interest is the generalization of the linear mapping between latents and the target, especially in the context of gene expression data. The regulatory activity of transcription factors on downstream genes can be non-linear. Methods developed by Behr et al. (2022; 2024) identify interacting pairs of genes with the assumption that the underlying genetic regulation can be modelled as a set of thresholded Boolean interactions between gene expression values. Our experimentation in 4.1 considered approximating this regulatory dynamic in $\mathbf{X} \rightarrow \mathbf{Z}$, where $\Phi(\mathbf{x})$ can approximate "spiky" regulatory functions due to our use of Relu activations. However, generalizing our identification results to permit non-linear, non-smooth $\mathbf{Z} \rightarrow \mathbf{Y}$ would strengthen the use of TBR for the study of gene regulatory relationships.

## Software and Data

Relevant code for replicating experimentation is released on Github. Gene expression data can be retrieved via the GTEx portal website.

### Acknowledgments

We acknowledge the support of the Natural Sciences and Engineering Research Council of Canada (NSERC).

Nous remercions le Conseil de recherches en sciences naturelles et en génie du Canada (CRSNG) de son soutien.

The authors would like to thank Paul Bertin and Alex Tong for helpful discussions.

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

# A  Appendix

## A.1  Proof of Proposition 4.4

*Proof.* By Assumption 3.1, and the conditions on $\hat{Y}$ outlined in Proposition 4.4, we have

$$\mathbb{E}[Y|\mathbf{X}, e] = \hat{w}_e^\top \hat{\Psi}(\mathbf{X}) \tag{24}$$

$$w_e^\top \Psi(\mathbf{X}) = \hat{w}_e^\top \hat{\Psi}(\mathbf{X}) \tag{25}$$

Let $e_1, \ldots, e_k$ be the environments given by Assumption 3.2. We can define matrices

$$\mathbf{U} = \begin{bmatrix} w_{e_1}^\top \\ \vdots \\ w_{e_k}^\top \end{bmatrix} \text{ and } \hat{\mathbf{U}} = \begin{bmatrix} \hat{w}_{e_1}^\top \\ \vdots \\ \hat{w}_{e_k}^\top \end{bmatrix}.$$

By Assumption 3.2, we know $\mathbf{U}$ is invertible, which allows us to write

$$\mathbf{U}\Psi(x) = \hat{\mathbf{U}}\hat{\Psi}(x) \tag{26}$$

$$\Psi(x) = \mathbf{U}^{-1}\hat{\mathbf{U}}\hat{\Psi}(x) \tag{27}$$

$$\Psi(x) = \mathbf{L}\hat{\Psi}(x) \tag{28}$$

where $\mathbf{L} := \mathbf{U}^{-1}\hat{\mathbf{U}}$.

By Assumption 3.3, we can create an invertible matrix $\mathbf{Q} = [\Psi(x^1), ..., \Psi(x^k)]$. We can subsequently denote $\hat{\mathbf{Q}} = [\hat{\Psi}(x^1), ..., \hat{\Psi}(x^k)]$, allowing us to write $\mathbf{Q} = \mathbf{L}\hat{\mathbf{Q}}$. Since $\mathbf{Q}$ is invertible, both $\mathbf{L}$ and $\hat{\mathbf{Q}}$ must also be invertible.

Combining Eq. 25 and Eq. 28 yields

$$w_e^\top \mathbf{L}\hat{\Psi}(x) = \hat{w}_e^\top \hat{\Psi}(x) \tag{29}$$

$$w_e^\top \mathbf{L}\hat{Q} = \hat{w}_e^\top \hat{Q} \tag{30}$$

$$w_e^\top \mathbf{L} = \hat{w}_e^\top , \tag{31}$$

where the last equation holds because $\hat{\mathbf{Q}}$ is invertible.

Choose some $a \in \mathcal{A}$ and assume $a = (e_0, e_0')$. Since Eq. 31 holds for all $e$, it holds in particular for $e_0$ and $e_0'$:

$$w_{e_0}^\top \mathbf{L} = \hat{w}_{e_0}^\top \tag{32}$$

$$w_{e_0'}^\top \mathbf{L} = \hat{w}_{e_0'}^\top . \tag{33}$$

By substracting both equations above, we obtain

$$(w_{e_0'} - w_{e_0})^\top \mathbf{L} = (\hat{w}_{e_0'} - \hat{w}_{e_0})^\top \tag{34}$$

$$\delta_a^\top \mathbf{L} = \hat{\delta}_a^\top , \tag{35}$$

where the last step above follows from the definition of $w_e$ in Eq. 2. This concludes the proof. $\qquad\square$

## A.2 Useful Lemmas

The argument for proving the following Lemma is taken from Lachapelle et al. (2022b).

**Lemma A.1** (Sparsity pattern of an invertible matrix contains a permutation)**.** *Let $L \in \mathbb{R}^{m \times m}$ be an invertible matrix. Then, there exists a permutation $\pi$ such that $L_{i,\pi(i)} \neq 0$ for all $i$.*

*Proof.* Since the matrix $L$ is invertible, its determinant is non-zero, i.e.

$$\det(L) := \sum_{\pi \in \mathfrak{S}_m} \mathrm{sign}(\pi) \prod_{i=1}^{m} L_{i,\pi(i)} \neq 0 , \tag{36}$$

where $\mathfrak{S}_m$ is the set of $m$-permutations. This equation implies that at least one term of the sum is non-zero, meaning there exists $\pi \in \mathfrak{S}_m$ such that for all $i \in [m]$, $L_{i,\pi(i)} \neq 0$. $\qquad\square$

## A.3 Remarks on § 3

### A.3.1 On relaxing the 1-sparse assumption:

Assumption 3.5 is particularly stringent, and many problem settings will not adhere to it in practice. We note some of the expected results of relaxing this assumption, as they relate to our theory.

Central to our proof are the two sets $\Gamma_i$ and $\Omega_i$. $\Omega_i$ is the set of indices where entanglement by $L$ decreases sparsity for column $i$, and $\Gamma_i$ is the set of indices where sparsity is increased. We reformulate the definition of $\Omega_i$

$$\Omega_i = \{ j \in S_{\Delta_{:,\pi(i)}} \mid \Delta_{j,-\pi(i)} \cdot L_{-\pi(i),i} = -\alpha \Delta_{j,\pi(i)} \} \tag{37}$$

One can see that the condition for index $j$ to be in $\Omega_i$ is equivalent to: $\Delta_{j,\pi(i)} \cdot L_{\pi(i),i} = 0$. Thus, if the number of indices in $\Omega_i = a$ for some $a \geq 2$, it implies that there is a subset of $a$ rows in $\Delta$ where the columns are linearly dependent. Under the assumption that non-zero values of $\Delta$ are generated independently, it seems intuitive that the likelihood of finding $a$ rows with linearly dependent columns decreases rapidly as $a$ increases, and the results in Figure 3a give some evidence towards this possibility. We leave for future work the possibility for formalizing a probabilistic bound on the size of $a$. We also note that Lachapelle et al. find an identification result for latent variables that holds in a setting with infinitely many environments but more general assumptions about sparsity, which may have some connections to the probabilistic line of reasoning we outline.

An additional possible direction for future work towards generalizing our result is to investigate the use of "anchor environments", as used by Moran et al. to disentangle latent generative factors.

Table 1: Summary of hyper-parameter settings.

| Experiment / Dataset | Parameter name | Setting or range |
|---|---|---|
| Simulated | $\lambda$ | 0.001 |
| Simulated | Learning rate | 0.001 |
| Gene Expression | $\lambda$ | {0.0, 0.1, 0.01, 0.001, 0.0001} |
| Gene Expression | Learning rate | {0.001, 0.0001} |

### A.4 Additional experimental details

**Implementation** We instantiated $\hat{\Psi}$ as a neural network with two hidden layers (256 units, 64 units) and LeakyRelu activations. Performance was compared across two methods of estimating linear map $\hat{\Phi}$. First, a TBR implementation was used to directly estimate $\hat{w}_0$ and $\hat{\Delta}$ in order to predict $\hat{y}$. Regularization of $||\hat{\Delta}||_0$ was approximated by instead regularizing $||\hat{\Delta}||_1$. We compare against a baseline of training a single estimation of $\hat{w}$, without accounting for variation between environments.

In each experiment, $\hat{\Psi}$ and the relevant estimation of $\hat{\Phi}$ were jointly optimized with the Adam optimizer Kingma & Ba (2014). We trained models on 50% of the data. A validation set of 25% of the data was used to tune hyper-parameters (learning rate, $\lambda$ for TBR models), and the remaining 25% was held out as a test set for evaluation. All models were implemented in PyTorch Paszke et al. (2019).

**Hyper-parameters**

Hyper-parameters are summarized in Table 1. Gene expression experiments were tuned individually per choice of **Z** and phenotype from the range listed.

**List of house-keeping genes**

Sets of genes for inclusion in **Z** described in § 4.1 were selected from the list in Table 2:

**Description of error estimates**

All results presented with a mean value accompanied by a $\pm$ error estimate indicate standard deviation, computed with a call to numpy's `np.std()` command. Box plots, such as Figure 3a, include quantiles and outliers calculated by seaborn's `sns.catplot()` function with default settings. Error bars on bar plots, such as Figure 5a indicate the 0.95CI and are computed by the `errorbar` function included in `sns.catplot` with default settings.

**Compute requirements**

Experimentation on the fully simulated dataset was performed on a MacBook Pro with 18GB memory and an M3 Pro CPU. Time to generate a simulated phenotype with $K = 1$ and fit an instance of TBR was estimated at $42.98 \pm 13.40$ over 10 repetitions.

Experimentation on gene-expression data described in § 4.1 was performed on a shared compute server with a total of 755GB of ram and 96 CPU cores. Training leveraged shared use of a Quadro RTX 6000 GPU, with an average GPU memory usage of approximately 1GB. Run-time to fit an instance of TBR for a random choice of **Z** was estimated at $25.37 \pm 2.53$ seconds over 10 repetitions.

### A.5 Additional experiments and results

Table 2: House-keeping genes

| Gene Name |
| --- |
| C1orf43 |
| CHMP2A |
| EMC7 |
| PSMB2 |
| PSMB4 |
| REEP5 |
| SNRPD3 |
| VCP |
| VPS29 |

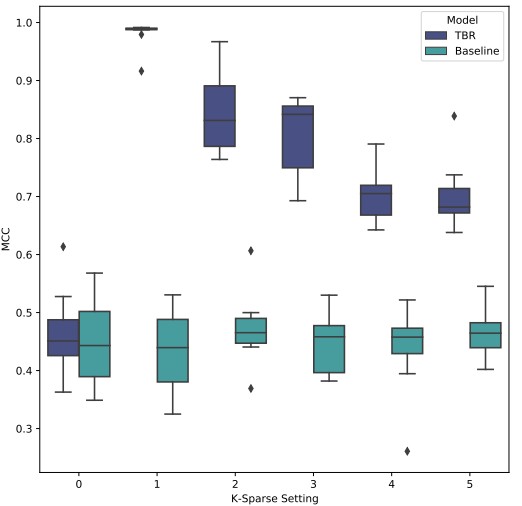

(a) Disentanglement performance (MCC).

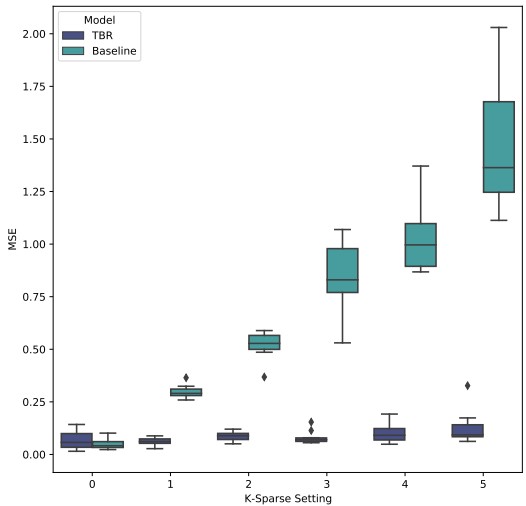

(b) Prediction error (MSE).

Figure 6: Assessment of model performance across various simulation settings, with observations available only that the leaf nodes.

**Simulation with observations exclusively at leaves**

Complementary to the experimentation depicted in Figure 3, we performed an equivalent set of experiments with the simulation altered such that only leaf nodes had observations, which more closely matches our motivating biological applications. We increased the depth of the tree from 7 to 8, keeping the number of observable environments similar. The results display the same trends as discussed in the main text, with an MCC approaching 1.0 exhibited by TBR in the $K = 1$ setting, and a significant increase in entanglement observed by both methods when $K = 0$.

**Simulation with more complex $X \to Z$ relationship** .

We further tested the robustness of the TBR method when increasing the complexity of the function relating $X$ to $Z$, by increasing the number of **tanh** layers from 1 to 2. MCC scores were slightly reduced across both methods for all settings, but MCC scores remained above 0.9 for TBR in the $S = 1$ setting. Results are depicted in Figure 7.

**Simulation with skewed distributions for $X$** .

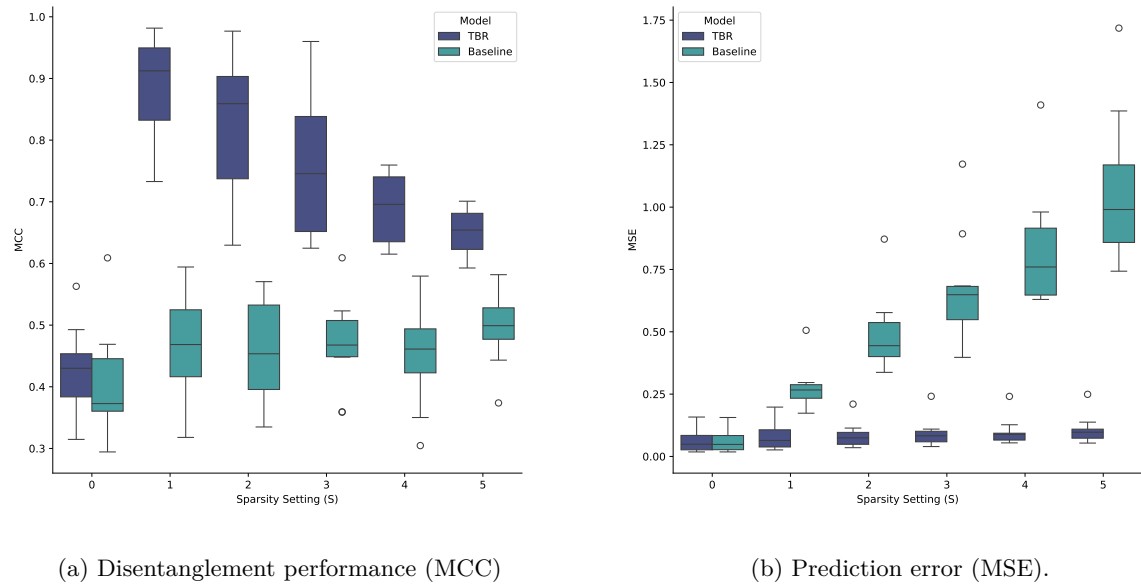

(a) Disentanglement performance (MCC)

(b) Prediction error (MSE).

Figure 7: Assessment of model performance across various simulation settings with a more complex $X \to Z$ function.

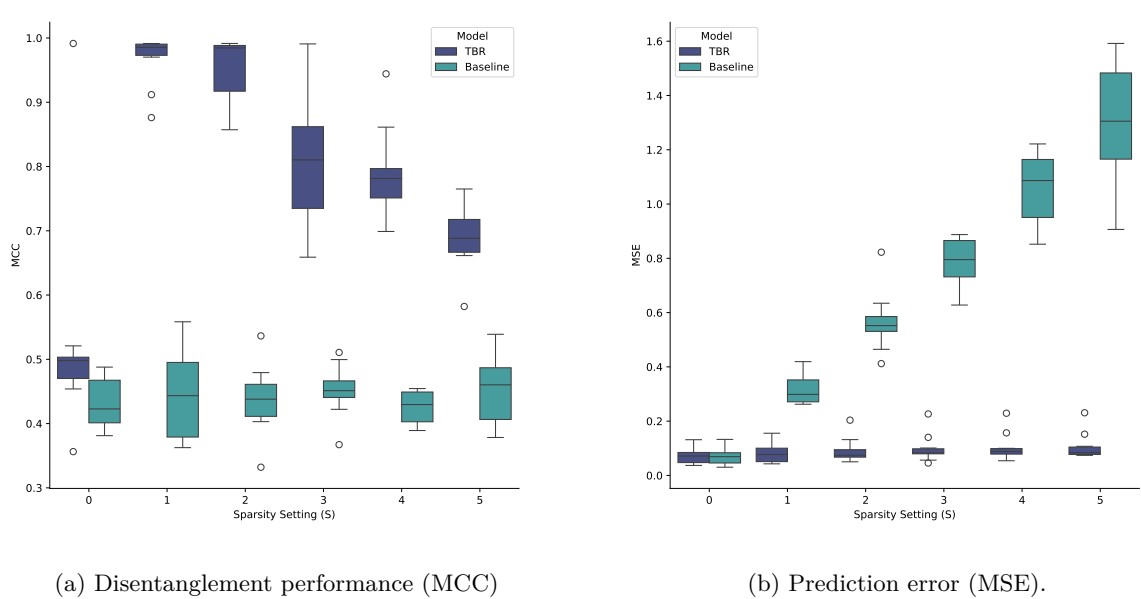

(a) Disentanglement performance (MCC)

(b) Prediction error (MSE).

Figure 8: Assessment of model performance across various simulation settings with non-Gaussian distributions of $X$.

We further tested the effects of considering alternative distribution for $X$ in our simulation studies, by distributing each $X_i$ as Gaussian with a skew value drawn uniformly from $(-1.0, 1.0)$. Our simulation results appeared robust to this modification. Results are depicted in Figure 8.

**Causal effect estimation in simulation experiments.**

Table 3: Mean-squared error of effect size estimates calculated using $\hat{\mathbf{Z}}$, in comparison to estimates calculated using ground-truth $\mathbf{Z}$, computed over 10 trials per environment. TBR achieves a much smaller mean error across all environments, showing that disentangled estimates of $\hat{\mathbf{Z}}$ allow for increased accuracy in determining the effect sizes of latent features.

| Environment | TBR MSE | Baseline MSE |
|---|---|---|
| Env 1 | $0.05 \pm 0.07$ | $0.87 \pm 0.76$ |
| Env 2 | $0.05 \pm 0.07$ | $1.08 \pm 0.91$ |
| Env 3 | $0.03 \pm 0.04$ | $1.29 \pm 1.67$ |
| Env 4 | $0.06 \pm 0.09$ | $1.68 \pm 1.72$ |
| Env 5 | $0.05 \pm 0.08$ | $1.96 \pm 3.58$ |
| Env 6 | $0.06 \pm 0.09$ | $2.55 \pm 2.14$ |
| Env 7 | $0.05 \pm 0.08$ | $1.3 \pm 0.94$ |
| Env 8 | $0.05 \pm 0.07$ | $1.85 \pm 1.3$ |
| Env 9 | $0.04 \pm 0.06$ | $1.19 \pm 1.11$ |
| Env 10 | $0.04 \pm 0.06$ | $1.3 \pm 1.29$ |
| **Pooled** | $0.05 \pm 0.07$ | $1.51 \pm 1.8$ |

The results from estimating causal effects from the estimated $\hat{\mathbf{Z}}$ using either TBR or the baseline method, as described in § 4, are summarized in Table 3.

**All-linear data-generative process**

We tested the effect of an all-linear DGP, where $\Psi(X)$ was replaced with a linear map with random weights. We maintained the neural network architecture of $\hat{\Psi}$. We noticed minimal change in the MCC performance of TBR, results are included in Figure 9.

**Stochastic Deltas**

Additionally, we tested the effects of randomly selecting the number of non-zero values in each $\delta_a$ by drawing from a Bernoulli distribution, while varying the parameter $\pi$ in order to control the expected level of sparsity. Results are displayed in Figure 10.

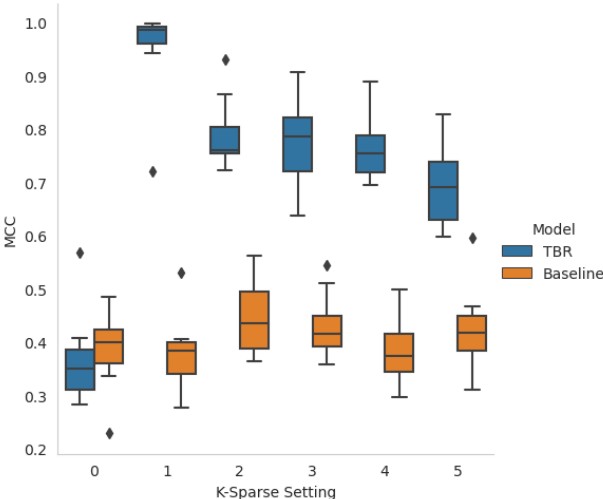

Figure 9: An equivalent plot to Figure 6a, obtained by using a linear map in place of $\Psi(\mathbf{X})$.

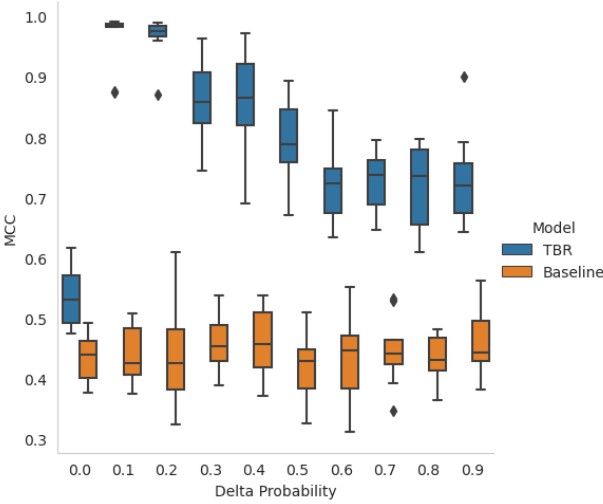

Figure 10: An equivalent plot to Figure 6a, obtained by using a Bernoulli distribution of non-zero entries in $\Delta$ in place of a fixed $k$-sparse setting.

