# OpenReview forum: "Sparsity regularization via tree-structured environments for disentangled representations"
_TMLR — Accepted by TMLR_

### Review · Reviewer_4mqX · 2025-03-11

**Summary Of Contributions:**

This paper studies learning disentangled representations using data from multiple environments and tasks. The focus is especially on the biological processes, for which multiple datasets (environments) are related by sparse changes and can be expressed in a hierarchical tree structure. For this setting, under a number of assumptions (most importantly, 1-sparse perturbations and linear mapping from latents to target), the proposed tree-based regularization is shown to help identify the latent variables. In simulations, the proposed method is tested on both synthetic data (which is a bit toy-ish) and gene expression data. Overall, the empirical results show promise that the proposed regularization technique can be helpful in practice.

**Audience:**

Yes

**Claims And Evidence:**

Yes

**Requested Changes:**

### **Organization and writing related points**

- Page 1. "*The challenge is in finding assumptions that are strong enough to rule out spurious solutions, while remaining flexible enough to fit the domain of interest*"
	I think this sentence should be more comprehensive. Currently, it only explains the *model* side, that is assumptions on the data generating processes. However, there is also *data* side (non i.i.d data, additional structures/information on the data etc.) as the paper leverages via sparse changes. I guess the authors mean to include everything when they say ``finding assumptions’’, but it’s not clear enough.

- Page 2. ""*These constraints reflect biological settings, where it is common to observe data across multiple cell types or model organisms with known relationships..*""
	Could you clarify what you refer to by known relationships, is it the hierarchical structure?
- Section 4 first paragraph: When describing the baseline method, why not make it more explicit (representations learned with and without regularization, with eq. x vs. eq .y, entangled vs. disentangled etc.)?

### **Discussion of assumptions and results**
- It seems like latent dimension $k$ needs to be known. Please clarify it.
- **Assumption 3.2 (sufficient task variability)**: This assumption is intuitive. However, I have two questions. First, is it possible to provide a counter-example to show that violating this assumption renders identifiability impossible? Second, while the assumption is intuitive, I think it should be made clearer earlier in the paper, at least via some hints. If I’m not mistaken, while the intro discusses the results, there is no mention of the assumption, and the reader has no idea until this point how identifiability will be possible (under tree-based regularization).
- **Assumption 3.3 (sufficient representation variability)**: Why not have a comment/explanation about the role of this assumption? I guess it’s a quite mild assumption, e.g., it is weaker than the full latent support assumption used in many recent CRL works.
- **Assumption 3.5 (1-sparse perturbations)**: Again, not sufficient interpretation on the role of this assumption? For instance, it seems almost analogous to single-node interventions used in recent multi-environment (interventional) CRL works (and even before that, sparse perturbations in Brehmer et al., etc.), where the perturbations are happening at a different level here.
- **Assumption 3.6 (sufficient perturbations)**: Again, it seems analogous to ``each node should be intervened/perturbed’’ condition. This totally makes sense, but the broader audience may not see the implications and necessity.
- **Proof of Proposition 3.7**: If the proof is given in the main body, I suggest moving it to the right after the proof sketch. Before giving the proof, I also would like to see a more detailed discussion of Proposition 3.7, which is the main result of the paper.
- Could you provide more intuition on whether the proposed tree-based regularization can be used for nonlinear mapping from latents to target variable? It does not have to be an affirmative answer, though understanding why or why not it can be extended would be useful.

### **Simulations**
- **Synthetic data setting**:  The considered setting is a bit too limited — only one setting of latent and observed dimension, only a simple nonlinear transform ($\Phi(x) = \tanh(w^T x)$). I suggest demonstrating the performance on at least some of the following settings: i) increasing dimension of $X$, ii) making the data less normalized, instead of using $N(0,1)$ for all $X$ variables, iii) considering a more complex nonlinear mapping from $X$ to $Z$, perhaps multi-layers of tanh.
- At the end of page 10, generalization to unseen environments is discussed. Can you explain this a bit more with respect to your modeling? For instance, is there a connection between holding out an environment (making it unseen) and violating an assumption (perhaps assumption 3.2)?

### **Related work**
- I suggest adding a couple more recent references for learning (causal) disentangled representations, e.g. for multiple environments(interventions) and multi-view CRL papers.
- “We consider latent variables which are causal descendants of the observations”. While this is clear in problem formulation, the distinction from the previous works should be made clearer earlier in the paper.
- Sparse changes to latents paragraph: If I’m not mistaken, the references in this paragraph are more on *paired samples*. As I mentioned above, there are also more recent interventional CLR studies that consider paired *distributions* instead.
- I’m not familiar with Layne et al. (2020). If the regularization scheme is indeed similar, I feel like it warrants a more detailed comparison.

Overall, I think the related work section need to be expanded, and perhaps giving it earlier in the paper (after introduction, before problem formulation).

### **Other minor comments**
- Page 1. Perhaps consider breaking the last sentence of the second paragraph into two sentences to improve readability.
- Figure 2. Might be better to give the reference just after Eq. (2) instead of at the end of Section 2.
- Proposition 3.4. Typo: equation equation.
- Please make sure to add appropriate hyperrefs, some are missing (for instance, Appendix A.1 after proposition 3.4)
- Figure 3 should come before Figure 4, according to discussion order.
- I suggest moving Table 3 to the main body, while still leaving the details in the appendix.

**Strengths And Weaknesses:**

**Data-generation process and objectives.**

The paper considers the following data generation process:
$$\mathbf{Z} = \Psi(\mathbf{X}) + \eta \ ; \quad Y^e = w_e^{\top}\mathbf{Z}^e + \epsilon \ , $$

where $\mathbf{X}$: observed variables, $\mathbf{Z}$: latent variables, and $Y$: target variable of interest.
This formulation (observed variables *cause* the latents) deviates from the standard CRL approaches. Though this is not necessarily a weakness, perhaps even can be considered as a strength if it’s able to explain the biological processes better (along with the hierarchical environment structure).

However, I want to emphasize two concerns.
1.  The linear mapping from latent $\mathbf{Z}$ to target variable $Y$ can be quite restrictive (the authors acknowledge this on page 2, with some motivation for nonlinear maps). That being said, I understand that studying linear maps is a reasonable starting point.
2. Latent variables $\mathbf{Z}$ are repeatedly referred as causal variables — since they *cause* target variable $Y$. However, reversing the data generation process ($X \rightarrow Z$, as opposed to $Z \rightarrow X$) also results in losing further causal interpretation of $Z$, e.g., there is no SCM interpretation to explain and identify the relationships among $Z_k$ variables.

**Sparsity regularization**: The proposed sparsity regularization is quite simple yet seems to be effective. I like this simplicity as a strength.

**1-sparse perturbations assumption**: Assumption 3.5 states that all changes in the hierarchical evolution of the latent processes are 1-sparse perturbations. The authors acknowledge this limitation.  Despite this weakness (limitation), simulations show that the proposed tree-based regularization is still helpful when the assumption is violated. I find this observation as one one the main strengths of the proposed method.

**Simulations**: While observations of the simulations are strong (for instance, the partial robustness against assumption violations as mentioned above and applicability to gene expression data), the synthetic data setting is quite limited. See my further comments in the requested changes.

**Insufficient explanations and organization of the paper**: Overall, the paper’s writing is good enough. However, at times, it leaves the reader yearning for more explanations (e.g., discussion of assumptions). This is not a major problem, but I have a number of suggestions/requests below.

---

> ### Author Response · Authors · 2025-04-24
> **Response to Reviewer 4mqX**
>
> We thank the reviewer for their time and considerable efforts spent considering our manuscript, and for their detailed comments and feedback. We describe below our responses to their listed concerns and modifications made with respect to their requested changes.
>
> **Organization and writing related points**
>
> >Page 1.''The challenge is in finding assumptions that are strong enough to rule out spurious solutions, while remaining flexible enough to fit the domain of interest" I think this sentence should be more comprehensive. Currently, it only explains the model side, that is assumptions on the data generating processes. However, there is also data side (non i.i.d data, additional structures/information on the data etc.) as the paper leverages via sparse changes. I guess the authors mean to include everything when they say ''finding assumptions’’, but it’s not clear enough.
>
> We ask the reviewer to clarify the distinction they hope to draw between ``assumptions on the data-generating process'' versus assumptions on the data -- we don't perceive the difference between the two. In the meantime, we clarify what we meant on page 1: we need sufficient assumptions about the data-generating process $P(X,Y)$ so that sparsity regularization (on the model side) when modeling this distribution suffices to disentangle features. In this paper, we establish the novel assumptions of sparsely varying mechanisms in the multi-task setting (i.e., assumptions on $P(X^e, Y^e)$, and show that sparsity regularization suffices to disentangle features in such data (under other assumptions about sufficient variability).
>
> >Page 2. ''These constraints reflect biological settings, where it is common to observe data across multiple cell types or model organisms with known relationships..'' Could you clarify what you refer to by known relationships, is it the hierarchical structure?
>
> Yes, ''known relationships'' refers to the hierarchical structures. We have clarified this in the text at line 48.
>
> >Section 4 first paragraph: When describing the baseline method, why not make it more explicit (representations learned with and without regularization, with eq. x vs. eq .y, entangled vs. disentangled etc.)?
>
> We have added some additional details and an equation denoting the objective function for the baseline method in the main text (Equation 21).
>
> **Discussion of assumptions and results**
>
> >It seems like latent dimension $k$ needs to be known. Please clarify it.
>
> Yes, $k$ needs to be known. This is implicit in the proof and discussed in the results section (see Figure 4), but we have now further clarified this assumption at line 108.
>
> > **Assumption 3.2 (sufficient task variability)**: This assumption is intuitive. However, I have two questions. First, is it possible to provide a counter-example to show that violating this assumption renders identifiability impossible? Second, while the assumption is intuitive, I think it should be made clearer earlier in the paper, at least via some hints. If I’m not mistaken, while the intro discusses the results, there is no mention of the assumption, and the reader has no idea until this point how identifiability will be possible (under tree-based regularization).
>
> While Assumption 3.2 is directly referenced in the proof of Proposition 3.4, it is also quite intuitive how sufficient task variability is necessary for the proof of Proposition 3.7. With an insufficient number of tasks exhibiting sparse effect changes, thr ground truth $\Delta$ would have all-zero columns. Clearly, these columns could be linearly entangled in $\hat{\Delta}$ at no increase in regularization cost, as they would remain fully sparse.
>
> We have added an earlier hint to this assumption at line 52.
>
> > **Assumption 3.3 (sufficient representation variability)**: Why not have a comment/explanation about the role of this assumption? I guess it’s a quite mild assumption, e.g., it is weaker than the full latent support assumption used in many recent CRL works.
>
> Assumption 3.3 plays a very similar role to Assumption 3.2 in the proof of Proposition 3.4 (showing that $\hat{Z}$ can be at most linearly entangled while admitting optimal predictions of $Y$. The proof technique followed is perhaps not so intuitive in terms of a connection to biology, but rather leverages the invertibility of the specified matrices to show that estimates of tasks $\hat{w}_e$ must be equivalent to ground truth tasks $w_e$ multiplied by some linear matrix. It is similar in technique to other proofs in CRL, in particular the 4 references listed on line 132 in the main text.

---

> ### Author Response · Authors · 2025-04-24
> **Response Continued**
>
> > **Assumption 3.5 (1-sparse perturbations)**: Again, not sufficient interpretation on the role of this assumption? For instance, it seems almost analogous to single-node interventions used in recent multi-environment (interventional) CRL works (and even before that, sparse perturbations in Brehmer et al., etc.), where the perturbations are happening at a different level here.
>
> Interventional CRL learns from multi-environment observations $P^e(X)$ and crucially leverages sparse changes across per-environment $P^e(Z)$ that arise from atomic interventions on a latent variable. In contrast, in the multi-task setting we consider, we observe $P^e(X, Y)$ and observe sparse changes across $P^e(Y|Z)$, where $Z$ is a latent mediating feature extracted from $X$. Specifically, the causal mechanism $P^e(Y|Z)$ _only varies along a single dimension_ of $Z$. While leveraging sparse perturbations is common in CRL, we stress that the sparsely varying causal mechanism setting is novel to CRL, and as we show in Figure 5b, has implications for zero-shot transfer learning.
>
> With regards to the interpretation of Assumption 3.5, we note that there is substantial discussion of the intuition and role of this assumption both within the proof sketch and immediately prior to Section 3.1.
>
> > **Assumption 3.6 (sufficient perturbations):** Again, it seems analogous to ``each node should be intervened/perturbed’’ condition. This totally makes sense, but the broader audience may not see the implications and necessity.
>
> We stress again that sufficient perturbations about the tasks $Y^e$ varying in sufficiently different ways from one another, based on changing the coefficients $W^e$, \emph{and not based on changing the distribution $P^e(Z)$}, which is what intervening on nodes refers to.
>
> > **Proof of Proposition 3.7:** If the proof is given in the main body, I suggest moving it to the right after the proof sketch. Before giving the proof, I also would like to see a more detailed discussion of Proposition 3.7, which is the main result of the paper.
>
> The proof for Proposition 3.7 is located immediately after the proof sketch. We note that both the opening paragraphs of Section 3 and the proof sketch discuss the intuition and utility of Proposition 3.7. We are happy to discuss specific aspects in more detail, but would appreciate the reviewer clarifying what aspects should be expanded upon.
>
> >Could you provide more intuition on whether the proposed tree-based regularization can be used for nonlinear mapping from latents to target variable? It does not have to be an affirmative answer, though understanding why or why not it can be extended would be useful.
>
> With linear mappings, we can linearly represent the changes across environments with $\Delta$. With nonlinear functions, formalizing sparse changes isn't straightforward. Moreover, when we consider only linear mappings from $Z \rightarrow Y$, the only indeterminacy in feature extraction $\hat{Z} = \phi(X)$ are invertible linear transformations, which we can rule out by regularizing the L0 norm of the linear shifts $\Delta$ that make up the mapping from $\hat{Z} \rightarrow Y$. If we considered nonlinear mappings, we have to rule out all nonlinear bijections of the disentangled features Z, requiring more assumptions.
>
> **Simulations**
> >I suggest demonstrating the performance on at least some of the following settings: i) increasing dimension of
> $X$, ii) making the data less normalized iii) considering a more complex nonlinear mapping from $X$ to $Z$, perhaps multi-layers of tanh.
>
> We have added new results to the Appendix implementing suggestion 2 - with random skew added to the distribution of each $X_i$ - and suggestion 3 - with two tanh layers in the $X$ to $Z$ relationship. In both cases, the disentanglement trends presented in the original results persist.
>
> **Related Work**
>
> >I suggest adding a couple more recent references for learning (causal) disentangled representations, e.g. for multiple environments(interventions) and multi-view CRL papers.
>
> We have added some additional references to the Related Work section, discussed starting at line 352.
>
> >''We consider latent variables which are causal descendants of the observations''. While this is clear in problem formulation, the distinction from the previous works should be made clearer earlier in the paper.
>
> We have clarified this distinction at line 62.

---

> > ### Author Response · Authors · 2025-04-24
> > **Response pt 3**
> >
> > >I’m not familiar with Layne et al. (2020). If the regularization scheme is indeed similar, I feel like it warrants a more detailed comparison.
> >
> > Layne et al. (2020) considered a similar tree-based regularization scheme for the task of improving predictive performance on biological data in the context of distributional shifts of the features or labels. Our implementation of TBR could be considered a special case of their regularization scheme utilizing L1 regularization (to approximate an L0 objective), but in contrast our work explicitly considers sparse changes in the mechanisms relating latent features to $Y$, and utilizes this objective to guarantee disentanglement. We have expanded the comparison between their method and ours at line 367.
> >
> > > Overall, I think the related work section need to be expanded, and perhaps giving it earlier in the paper (after introduction, before problem formulation).
> >
> > Some additional references have been added to the Related Work section. Our preference is to leave the section at it's current location, which we feel is better for the flow of the paper. We could consider moving it for the final version if it is strongly felt that this improves the manuscript.
> >
> > **Other minor comments**
> >
> > We appreciate these suggestions, and they are all now reflected in the text, with the exception of the final suggestion - we opt to leave Table 3 in the Appendix for conciseness, as the overall results are already described in the main text.

---

### Review · Reviewer_FWx2 · 2025-03-18

**Summary Of Contributions:**

The authors proposed a tree-based penalization to recover disentangled representations. The authors established identifiability results, and showed sparse penalization lead to disentangled representations.

**Audience:**

Yes

**Broader Impact Concerns:**

N/A.

**Claims And Evidence:**

Yes

**Requested Changes:**

- I think the author(s) had to put some assumptions on the true $\Psi$ and the neural net to show the *entire* procedure to identify and it cannot be a general neural net (e.g., a constant function is a weird neural net). I would encourage the authors to spell that out. It can even take the form that $\hat{\Psi}$ can approximate $\Psi$ well in some clearly stated standard. Similarly, what is the assumption on $\lambda$?

- It is a bit hidden in the appendix that instead of optimizing the $L_0$ penalized objective (4), the author instead optimized
\begin{equation}
\sum_{e\in \mathcal{L}}(w_e^\top \hat{\Psi}_{\theta}(\mathbf{X}^{e})-Y^e)^2+\lambda||\hat{\Delta}||_1
\end{equation}
This is a significant relaxation that must be transparent in the main text. I encourage the authors to make some comments on this relaxation. It will not only encourage sparsity but also shrink values of other non-0 elements in $\hat{\Delta}$ towards 0, a behavior that does not exist when penalizing with $||\hat{\Delta}||_0$. Also, optimizing this is not that trivial: e.g., one cannot use naive gradient descent since the gradient will not exist at the sparse solution, like LASSO. In the paper, the authors just stated they implemented this in `pytorch` without details.

- Choice of the penalization strength $\lambda$ is not quite transparent, in appendix A4 the authors only stated they are tuned from the list but no details on how. Maybe I missed the detail.

- In assumptions 3.1-3.2, does the invertibility also imply some dimension assumption?

**Strengths And Weaknesses:**

**Strengths**: Reasonable theoretical contributions, and well-performing algorithms.

**Weaknesses**: the algorithm is not very connected to the theory.

---

> ### Author Response · Authors · 2025-04-24
> **Response to Reviewer FWx2**
>
> We thank the reviewer for their time spent considering our manuscript and for their contributions. We will address their concerns below:
>
> > *I think the author(s) had to put some assumptions on the true $\Psi$ and the neural net to show the entire procedure to identify and it cannot be a general neural net (e.g., a constant function is a weird neural net). I would encourage the authors to spell that out. It can even take the form that $\hat{\Psi}$ can approximate well $\Psi$ in some clearly stated standard. Similarly, what is the assumption on $\lambda$?*
>
> We consider this concern to be a misunderstanding of the problem formulation. No claim is made that $\hat{\Psi}$ can identify the specific functional form of $\Psi$, but simply that the neural network $\hat{\Psi}$ can approximate $\Psi$. This is equivalent to stating that $\Psi$ is a continuous function subject to the Universal Approximation Theorem for neural networks. We have clarified this in the main text at line 67.
>
> > *It is a bit hidden in the appendix that instead of optimizing the penalized objective (4), the author instead optimized*
> $$
> \sum_{e \in \mathcal{L}} w_{e} \top \hat{\Psi_{\theta}}(X^{e}- Y^{e})^{2} + \lambda || \hat{\Delta}|| \small{1}
> $$
> >    *This is a significant relaxation that must be transparent in the main text. I encourage the authors to make some comments on this relaxation. It will not only encourage sparsity but also shrink values of other non-0 elements in $\hat{\Delta}$ towards 0, a behavior that does not exist when penalizing with $|| \hat{\Delta} ||_{0}$. Also, optimizing this is not that trivial: e.g., one cannot use naive gradient descent since the gradient will not exist at the sparse solution, like LASSO. In the paper, the authors just stated they implemented this in pytorch without details.*
>
> We appreciate the reviewer's suggestion that the use of $L_1$ regularization should be specified within the main text, and have moved this fact from the Appendix line 208 of the main text. We also acknowledge that this regularization scheme will also shrink non-zero values in $\hat{\Delta}$, and have added a line in the main text acknowledging this minor discrepancy between our theory and implementation. However, we disagree that optimizing this with pytorch is non-trivial. The gradient of $L_1$ penalties is smoothly differentiable at all values other than the point discontinuity at 0, and it is extremely common in machine learning to approximate sparse solutions with a combination of $L_1$ regularization terms and standard gradient-descent optimization (or common alternative such as Adam). Notably, this approximation of L0 with L1 has been successfully used by other CRL works (see [1,2]).  Indeed, a soft-thresholding operator could be applied to $\hat{\Delta}$ to produce truly sparse solutions, but we do not expect that this would affect the ability of TBR to learn disentangled representations.
>
> [1] Lachapelle, S., Rodriguez, P., Sharma, Y., Everett, K. E., Le Priol, R., Lacoste, A., \& Lacoste-Julien, S. (2022, June). Disentanglement via mechanism sparsity regularization: A new principle for nonlinear ICA. In Conference on Causal Learning and Reasoning (pp. 428-484). PMLR.
>
>
> [2] Lachapelle, S., Deleu, T., Mahajan, D., Mitliagkas, I., Bengio, Y., Lacoste-Julien, S., \& Bertrand, Q. (2023, July). Synergies between disentanglement and sparsity: Generalization and identifiability in multi-task learning. In International Conference on Machine Learning (pp. 18171-18206). PMLR.

---

> > ### Author Response · Authors · 2025-04-24
> > **Response continued**
> >
> > > Choice of the penalization strength $\lambda$
> >  is not quite transparent, in appendix A4 the authors only stated they are tuned from the list but no details on how. Maybe I missed the detail.
> >
> > This is indeed specified within appendix A4. Across all experiments, $25\%$ of samples are reserved as a validation set. As in standard practice, predictive performance on the validation set is used to tune hyper-parameters, including $\lambda$.
> >
> > >In assumptions 3.1-3.2, does the invertibility also imply some dimension assumption?
> >
> > We assume that the reviewer is referring to Assumption 3.2 (sufficient task variability) and Assumption 3.3 (sufficient representation variability), which make reference to invertible matrices. In the case of Assumption 3.2, the dimension $k$ of the matrix is equal to the number of parameters in $w$, requiring that there be (at minimum) $|w|$ different tasks. In Assumption 3.3, the relevant dimension $k$ is equal to the dimension of $Z$, and there must be at minimum $|Z|$ samples in $X$ to select from.

---

### Review · Reviewer_dfRA · 2025-03-24

**Summary Of Contributions:**

This paper proposes a novel disentangled representation learning method for biological settings called Tree-Based Regularization (TBR). It infers latent variables from low-level observations across different environments (e.g., gene expressions from different cell types), and the learned latent causal variables can give valuable scientific insights. The proposed TBR method is validated by theoretical proofs and simulated experiments.

**Audience:**

Yes

**Broader Impact Concerns:**

After a review of the submitted work, no concerns regarding ethical implications have been identified.

**Claims And Evidence:**

Yes

**Requested Changes:**

1. Include the performance comparison of TBR and baseline approach on a real-world imbalanced dataset in the empirical studies.

2. Include more competitive baseline approaches for empirical studies in addition to the current baseline approach.

**Strengths And Weaknesses:**

Strengths:

1. The key assumption of TBR is a theoretically valid assumption. It assumes that only a sparse set of true latents will change across closely related environments. Therefore, TBR applies a sparsity-regularization term in the loss function to induce sparsity and recover true latent variables.

2. This paper proves that the regularization on the matrix $\hat{\Delta}$ ensures the identification of true latent variables up to permutation and element-wise rescaling, as only linear transformations can be applied to $\Delta$ while maintaining the same level of sparsity. Thus, TBR prevents the entanglement of the components of the true features.

3. This paper performed experiments both on simulated data and real data to validate the effectiveness of the proposed TBR. On simulated data, this paper proves the capability for disentanglement and causal effect estimation of TBR. The experiments on real data also proved better disentanglement than baseline approaches.

Weaknesses:
1. This paper subsampled the dataset of real data during experiments to ensure balanced cell types. While this experiment demonstrates the effectiveness of TBR on balanced data, real-world data are highly imbalanced. It is unclear whether TBR would be robust even on highly imbalanced data.

2. The authors could consider including more baseline approaches for empirical studies, such as the ones mentioned in the related work section. The current baseline approach might be too simple and TBR would almost certainly demonstrate improved performance.

3. The proofs of TBR are valid. But more feedback from reviewers with ML background is certainly welcome.

---

> ### Author Response · Authors · 2025-04-24
> **Response to Review dfRA**
>
> We thank the reviewer for their time and efforts considering our manuscript. We appreciate their comments noting the validity of our theoretical findings and the connections to real-world biological challenges.
>
> With regards to the requested changes:
>
> * **On the use of balanced vs imbalanced data:** we appreciate the reviewer's observation that many tasks of interest will require handling imbalanced classes.To ensure understanding, we clarify that TBR is robust to class imbalance in a categorical $Y$.  Our dataset of single-cell gene expression did feature a small number of environments (e.g. cell types) that were vastly over-represented. We under-sampled these environments to ensure that training did not involve an excessive number of batches with samples from only 1 or 2 environments, which would violate the assumption of sufficient task variability.
>
> * **On additional baselines:** We view a primary contribution of our work as establishing a novel set of assumptions, motivated by phylogeny, that permit the identification and subsequent disentanglement of latent variables. Other disentanglement methods will of course utilize different sets of assumptions, corresponding to alternative data generative processes, making a direct comparison challenging. The claim we make is thus not that TBR is the ''best disentanglement method", but rather that we have established a new space of generative processes where disentanglement is possible. That being said, in the spirit of strengthening our submission, we have added an additional comparison method to our main simulation result. We considered the ''Sparse Predictors'' method studied in [1], as it also considers disentangled feature learning in a multi-task setting. But crucially, unlike our assumptions about sparse mechanism changes across tasks, their work focuses on cases when tasks sparsely depend on features. We expect this assumption to be violated in our generative processes of interest, and indeed find that, with regard to disentanglement, TBR substantially outperforms ''Sparse Predictors'' in settings with sparse mechanistic changes, as is now depicted in Figure 3 in the main text.
>
> [1] Lachapelle, S., Rodriguez, P., Sharma, Y., Everett, K. E., Le Priol, R., Lacoste, A., \& Lacoste-Julien, S. (2022, June). Disentanglement via mechanism sparsity regularization: A new principle for nonlinear ICA. In Conference on Causal Learning and Reasoning (pp. 428-484). PMLR.

---

### Author Response · Authors · 2025-04-24
**Revision added**

Dear reviewers and editors,

We have uploaded a revised version of the manuscript incorporating suggested changes from the reviewers. Please find the modified text marked in red. Line numbers were temporarily added for ease of referral.

Sincerely,

Authors

---

### Author Response · Authors · 2025-06-13
**Anticipated timelines + next steps**

Dear AE,

We the authors thank you for your continued service with regards to our submission.

We wanted to confirm whether there are any further actions required from us at this time, and that our submission is on track for the next steps in the process.

With thanks,

The authors

---

### Author Response · Authors · 2025-07-17
**Camera Ready version submitted**

Dear AE and TMLR Editors in Chief,

We have now submitted the camera-ready version of our manuscript.

Many thanks again to the reviewers and action editor.

Sincerely,

The authors

---

### Decision · Action_Editor_Kgg6 · 2025-06-18

**Recommendation:** Accept with minor revision

**Additional Comments:**

Weakness:
   My main revision ask is related to some prior literature that is not cited/discussed that relates to phenotype prediction from transcription factors/gene expression. There is a line of work starting with this preprint https://arxiv.org/pdf/2102.11800 and ending with the following PLOS one publication https://journals.plos.org/plosone/article?id=10.1371/journal.pone.0298906. Authors are strongly encouraged to discuss the model in these papers and their approach. I do acknowledge that the above works do not consider multiple environments (that are related by trees). Certainly there is novelty in that respect.

 However, my main concern is regarding the representation function $\Psi(x)$ which creates a representation of gene expression $Z$ from transcription factors $X$ as in the real data experiments. The above papers argue that contrary to the neural net representation (which is usually smooth and Lipschitz) that the authors adopt, this representation function is actually *spiky*. The reason, they argue is, that biological models for phenotype exhibit thresholding behavior. Multiple transcription factors becoming higher together - this interaction results in a latent gene being expressed. So they have linear model of higher order thresholded boolean interactions.

So what may be warranted is a discussion and perhaps another simulation where the true phenotype is according to the linear spiky model with boolean interactions and if the proposed method does well in that case or not.

**Audience:**

Yes

**Audience Explanation:**

Finding common representations on top of which predictors vary mildly across environments is an important problem in general for multi task learning and this identifiability result (although simple) is novel and it seems very relevant to the biological application.

**Claims And Evidence:**

Yes

**Claims Explanation:**

Summary:
 Paper considers the problem of learning a common latent representation and a linear model to predict a target on top of it that sparsely varies across environments. Sparse variation across environments, motivated by phenotype prediction in genomics, is modeled as a tree where each environment descends from another symbolized by a very sparse variation in the linear model. Given data for all environments that is tree structured, authors show that sparsity penalty on the linear model variations and a Neural net parameterization of the representation can uniquely learn the representation under various assumptions. Paper then shows some simulations showing recovery , ablations when assumptions are violated and finally results on a genomics dataset.

Overall:
I read one of the proofs myself and it seems like the central claims are correct. Simulations and then real data experiments reflect the efficacy of the method. Reviewers also tend to agree with this after the rebuttal period.

I have a concern regarding related work which I have quoted below. Please see that.